# Atg8-Family Proteins—Structural Features and Molecular Interactions in Autophagy and Beyond

**DOI:** 10.3390/cells9092008

**Published:** 2020-09-01

**Authors:** Nicole Wesch, Vladimir Kirkin, Vladimir V. Rogov

**Affiliations:** 1Institute of Biophysical Chemistry and Center for Biomolecular Magnetic Resonance, Goethe-University Frankfurt, 60438 Frankfurt am Main, Germany; Wesch@bpc.uni-frankfurt.de; 2Cancer Research UK Cancer Therapeutics Unit, The Institute of Cancer Research London, Sutton SM2 5NG, UK; vladimir.kirkin@icr.ac.uk; 3Structural Genomics Consortium, Buchmann Institute for Life Sciences, Goethe-University Frankfurt, 60438 Frankfurt am Main, Germany; 4Institute of Pharmaceutical Chemistry, Goethe-University Frankfurt, 60438 Frankfurt am Main, Germany

**Keywords:** Atg8, LC3, GABARAP, LIR motif, SAR, UBL, autophagy

## Abstract

Autophagy is a common name for a number of catabolic processes, which keep the cellular homeostasis by removing damaged and dysfunctional intracellular components. Impairment or misbalance of autophagy can lead to various diseases, such as neurodegeneration, infection diseases, and cancer. A central axis of autophagy is formed along the interactions of autophagy modifiers (Atg8-family proteins) with a variety of their cellular counter partners. Besides autophagy, Atg8-proteins participate in many other pathways, among which membrane trafficking and neuronal signaling are the most known. Despite the fact that autophagy modifiers are well-studied, as the small globular proteins show similarity to ubiquitin on a structural level, the mechanism of their interactions are still not completely understood. A thorough analysis and classification of all known mechanisms of Atg8-protein interactions could shed light on their functioning and connect the pathways involving Atg8-proteins. In this review, we present our views of the key features of the Atg8-proteins and describe the basic principles of their recognition and binding by interaction partners. We discuss affinity and selectivity of their interactions as well as provide perspectives for discovery of new Atg8-interacting proteins and therapeutic approaches to tackle major human diseases.

## 1. Introduction

Autophagy is a fundamental process of degradation and recycling of cellular components to maintain homeostasis [1,2]. In concert with, but also far beyond, the ubiquitin-proteasome degradation system (UPS), autophagy serves to remove bulky cytosolic cargo, such as long-lived protein and protein complexes, lipid droplets, portions of and whole organelles, cytosolic bacteria, etc., (reviewed in [3,4,5]). This is achieved by enclosing the cargo by a lipid membrane and subsequent trafficking into degradative cellular compartments—vacuoles (in fungi and plants) and lysosomes (other eukaryotes). The autophagy pathway branches into three relatively independent subtypes: *microautophagy*, in which the cytoplasmic cargo is directly engulfed by invagination of the lysosomal membrane; *chaperon-mediated autophagy*, which relies on translocation of unfolded proteins across the lysosome membrane in a receptor-depended manner, [1]; and *macroautophagy* (hereafter simply *autophagy*)*,* which makes use of a double-membrane vesicle (*autophagosome*) as a shuttle between the cytosol and the degradative compartment.

Autophagosomes are formed by multiple mechanisms (reviewed in [6]) from a primordial membrane (*phagophore* or *isolation membrane*) that elongates by adding additional lipid and engulfs cargo in either selective or non-specific manner. After expansion and closure of the membranes, in a so-called maturation step, autophagosomes fuse with the vacuoles/lysosomes, in which multiple hydrolytic enzymes digest the engulfed cargo and molecular building blocks can be released into the cytosol for recycling or downstream catabolic reactions.

This complex process is orchestrated by >36 evolutionary conserved autophagy-related (Atg) proteins. In 2016, the Nobel Prize in Physiology and Medicine was awarded to Yoshinori Ohsumi for his discovery of Atg genes in 1990s. Within Atgs, autophagy-related, ubiquitin-like modifiers (Atg8 in yeast, LC3/GABARAP proteins in mammals) play a special role. Like many other ubiquitin-like proteins (UBLs), Atg8/LC3/GABARAPs display a high structural similarity to ubiquitin and are expressed as inactive precursors that undergo proteolytic maturation (performed by Atg4 proteases) to expose their invariant *C*-terminal Gly residue participating in a ubiquitin-like substrate conjugation reaction, involving Atg7 (E1), Atg3 (E2), and Atg5~Atg12: Atg16L1 complex (E3) proteins [7,8,9]. Of note, Atg5 and Atg12 are themselves autophagy-related UBLs, covalently bound in a separate conjugation cascade involving Atg7 (E1) and Atg10 (E2) enzymes (reviewed in [4]). The concerted action of the UBL conjugation cascade results in the formation of a covalent bond between the *C*-terminal Gly of Atg8/LC3/GABARAPs and the amino group of the substrate lipid, phosphatidylethanolamine (PE). PE is the second most abundant phospholipid (the first one being phosphatidylcholine, PC) that participates in many cellular pathways (reviewed in [10]), populating among others endoplasmic reticulum (ER) membranes and autophagosomes. In their lipid-conjugated form, Atg8/LC3/GABARAPs become embedded in the autophagosomal inner and outer membranes [11].

Being the core Atg proteins, Atg8s and their conjugation machinery are often mutated in different experimental models to study the consequence of autophagy deficiency. *Atg8/Apg8/Aut7/Cvt5* is a non-essential gene in yeast; however, its deficiency leads to the defect in selective and non-selective forms of autophagy [12]. Prolonged stress conditions, such as nitrogen starvation, lead to decreased fitness of Atg8- (and autophagy-) deficient yeast cells [13]. Similarly, inactivation of two Atg8 homologues in *C. elegans*, *LGG-1* and *LGG-2*, leads to significant life span shortening in this organism [14]. Because of the gene duplication in Metazoans, instead of disrupting Atg8, knockout of Atg7 has been employed to study the physiological role of autophagy. Lack of Atg8 lipidation results in neurodegeneration in *Drosophila* [15] and mice [16]. Later studies, however, showed that Atg8-/Atg5-/Atg7-mediated autophagy is required for homeostasis of a majority of organ systems [17], while its disruption leads to premature aging in mice [18]. In higher plants, Atg8 and the conjugation machinery are required for resilience under adverse conditions [19]. Intriguingly, several forms of non-canonical autophagy have been described that can proceed (albeit inefficiently) without the functional Atg8 conjugation system [20,21,22].

Atg8/LC3/GABARAPs participate in all steps of the autophagosome biogenesis: phagophore initiation and elongation as well as autophagosome maturation and fusion with the vacuole/lysosome; they also mediate the *selectivity* of the autophagy pathway. *Selective autophagy* is an evolutionary adaptation of the bulk autophagy, aimed at sequestering and degrading specific types of cargo [23,24]. Here, the cargo is recognized by a growing number of *selective autophagy receptors* (SAR), which simultaneously bind the target and the components of the autophagosome, mainly Atg8-family proteins, triggering a chain of events resulting in cargo sequestration in the autophagosomes.

Most SARs display modular structural organization, possessing specific domains/motifs for saturated intermolecular network (reviewed in [5,25]). The size and domain content of different SARs could vary significantly; however, all known SARs seem to contain sites to mediate interactions with Atg8/LC3/GABARAPs. For instance, the SARs of the sequestosome-1-like receptor (SLR) group, such as the founding member p62/SQSTM1, interact with ubiquitin and LC3/GABARAPs, thereby mediating degradation of ubiquitin-decorated cargo, such as protein aggregates [5]. Interactions between Atg8/LC3/GABARAPs and SARs were extensively investigated in the past decade, resulting in the elegant and powerful concept of LC3-Interacting Region (LIR) motif (also known as Atg8-Interacting Motif, AIM, for the fungi and plants) [26,27,28].

The canonical LIR is a short (up to 10 residues) unstructured or β-stranded region within SARs (and other Atg8-interacting proteins) responsible for the efficient and selective recognition of the autophagosome-embedded Atg8/LC3/GABARAP proteins. In the case of the SARs, this interaction mediates tethering of the cargo-SAR complex to the autophagosome for the subsequent degradation. There are more than 50 structures of Atg8/LC3/GABARAPs proteins with various LIR motifs in the protein data bank (PDB), and the amount of the structural information on the Atg8/LC3/GABARAP interactions grows constantly. Based on this information, computational approaches to predict and validate LIR motifs within specific proteins of interest or within a given proteome were successfully established [29,30]. Additionally, a number of biochemical and biological methods to validate LIR motifs were developed and broadly used in research [31,32,33,34].

However, together with the obvious progress in elucidating the interactions between Atg8/LC3/GABARAPs and SARs, recent structural and functional studies revealed a number of fundamental insufficiencies in global understanding of the Atg8/LC3/GABARAPs cellular functions. For example, very little is known about the non-autophagic roles of the Atg8/LC3/GABARAPs. Also, presently it is not clear if the LIR motif is the only structural determinant mediating interaction between Atg8/LC3/GABARAPs and their interaction partners. What determines affinity and selectivity of these interactions and how can one modulate them? Is it possible to rationally modify the LIR motif in order to target a specific member of Atg8/LC3/GABARAP in vivo?

In this review, we reanalyzed available information on the Atg8-family proteins, including their specific differences in sequences, structure and functions, carefully and extensively reviewed in the past years [4,5,11,25,35,36,37,38]. However, the main goal of this work is to cast new light on the previously reported features of Atg8/LC3/GABARAP as well as to provide clarity on similarities vs. differences between the individual members of the subfamilies. We review known determinants of molecular recognition demonstrated and/or suggested for Atg8/LC3/GABARAP proteins as they perform their autophagy-related and autophagy-independent functions, but also indicate novel structural motifs that may be implicated in the Atg8/LC3/GABARAP interactions. We describe the current state with respect to the canonical and atypical LIR-motifs, their *N*- and *C*-terminal extensions and provide a structural view on the LIR-independent interactions, including helical binders and novel ubiquitin-interacting motif (UIM)-dependent (UDS)-binders. We also present a current view on the selectivity determinants (within Atg8/LC3/GABARAP proteins and LIR motifs) and expand on the potential use of highly affinitive and highly selective Atg8/LC3/GABARAP binders in research and therapy.

## 2. Ubiquitin, UBLs, and Atg8-Family Proteins

### 2.1. Structural Overview

Ubiquitin is a small globular protein of 76 residues, highly conserved from yeast to human. Its tightly folded structure, known as the “β-grasp fold,” is characterized by the five-stranded β-sheet wrapped around the central α-helix [39] (Figure 1A). Another evolutionarily conserved feature of ubiquitin is its synthesis as a precursor protein that undergoes proteolytic maturation to expose a *C*-terminal Gly residue, whose carboxyl group can be conjugated to an amino group of a Lys or an *N*-terminal residue of another ubiquitin or a great variety of other proteins. A cascade of enzymatic reactions, involving activating (E1), conjugating (E2), and ligating (E3) enzymes, generates ubiquitin conjugates containing either monoubiquitin or polyubiquitin chains, whose linkage and topology are stipulated by the choice of one of the seven internal ubiquitin Lys residues (Lys6, Lys11, Lys27, Lys29, Lys33, Lys48, and Lys63). Ubiquitin-binding domain (UBD)-containing proteins interact with conjugated ubiquitin in a non-covalent fashion and act as ubiquitin receptors, which can mediate assembly of structural or signaling complexes [40]. Importantly, the ubiquitin signal can be reversed through the action of a large class of deubiquitinating enzymes (DUBs), which are proteases capable of cleaving the isopeptide bond between ubiquitin and its substrate [41].

A superfamily of ubiquitin-like (UBL) modifiers share with its founding member the characteristic fold (but not necessarily the primary sequence) as well as the ability to become covalently conjugated to a substrate, which can be either a protein (in case of e.g., SUMO, Nedd8, UFM1, and Atg12) or a lipid (in case of Atg8/LC3/GABARAP). Like with ubiquitin, activity of dedicated proteases is required to produce a mature UBL as well as to deconjugate it from the substrate for signal termination [40]. Atg12 was the first UBL found in yeast as an essential autophagy protein [42]. Atg12 becomes conjugated to Lys130 of Atg5. Atg5 itself contains two UBL domains [43], so that the Atg12~Atg5 conjugate comprises three UBL moieties, which may be important for the recruitment of other factors required for phagophore elongation. There is growing evidence that Atg5 interacts with selective autophagy receptors (SARs), such as Atg19, optineurin, p62/SQSTM1, and NDP52 [44,45]. This interaction was proposed to be mediated by AIM/LIR and stimulate Atg8 conjugation. On the other hand, clathrin-mediated vesicular trafficking, including clathrin heavy and light chains, and several clathrin adaptors, have been identified in the Atg5/Atg12 interactome [46], and earlier studies identified clathrin heavy chain as a direct interactor for GABARAP [47]. Additional work is required to clarify the exact interaction surfaces between the UBLs in the Atg12-Atg5 conjugate and partner proteins.

Intriguingly, Atg12 can also be conjugated to other proteins [48]. However, the canonical autophagic role of the Atg12~Atg5 conjugate is in the complex with Atg16, which acts as an E3 ligase for Atg8-PE conjugation, as illustrated by experiments with ectopic expression of Atg16 [49].

The other autophagy-related UBL, Atg8, as well as its mammalian homologs, LC3 and GABARAP proteins, besides structural homology to ubiquitin reveal a significant similarity in their biogenesis. They are synthesized as precursors and undergo processing by Atg4 proteases. Mature Atg8, with the exposed *C*-terminal Gly, is activated by Atg7 (E1 enzyme), transferred to Atg3 (E2), and finally linked to the amino group of PE [52] via the Atg12~Atg5:Atg16 complex (E3). Members of the Atg8/LC3/GABARAP family are the only known UBLs that modify a lipid. Atg8-PE localizes on phagophores. Upon autophagosome maturation, Atg8 is deconjugated from the outer membrane by Atg4—a step important for the phagophore closure [53]; while Atg8 conjugated to the inner membrane of the autophagosome is delivered to the lysosome, where it is degraded together with the autophagosome and its contents. Since Atg8 (and also LC3) is associated with the autophagosome at all times throughout its biogenesis, it is used as a *bona fide* marker for autophagosome formation [1].

Structurally, the Atg8/LC3/GABARAP proteins form a family of GABARAP-like proteins within the “ubiquitin-like” superfamily (structural classification of proteins database, SCOP [54]). Similar to the ubiquitin, Atg8/LC3/GABARAP proteins possess a sheet of mixed parallel/antiparallel β-strands wrapped around a central α-helix and decorated with auxiliary α-, 3.10-helices and loops (Figure 1B).

Upon analysis of the available Atg8/LC3/GABARAP structures, we have noticed that non-processed LC3/GABARAP proteins (LC3A, LC3B, and GABRAP; no structural data is available for the non-processed LC3C; no α5 helix in Atg8) display in some structures an additional α-helix at their C-terminus (Figure 1C). This α-helix α5 is mostly associated with non-processed LC3/GABARAP proteins in complex with various LIR-motifs. It seems that the processed (and lipidated) forms of LC3/GABARAPs are not decorated with this accessorial α-helix; thus, we speculate that in the full length, non-processed forms of LC3 this α-helix could participate in some functional protein–protein interactions inside and outside of the autophagy pathway. However, existence of the helix α5 in different Atg8/LC3/GABARAP proteins, its stability in dependence of various factors, and its functional role(s) have to be thoroughly investigated. Despite low sequence correlations, Atg8, LC3, and GABARAP proteins show very high structural similarity—their structures could be overlaid upon each other with a root-mean-square deviation (RMSD) of 1.2 Å on backbone atoms (Figure 1D).

The main structural difference between Atg8/LC3/GABARAP proteins and other UBLs, which also determines the specific role of Atg8/LC3/GABARAP in autophagy, is the presence of two extra α-helices located *N*-terminally to the ubiquitin core. This *N*-terminal α-helical subdomain significantly varies in the amino-acid content among the different members of the Atg8/LC3/GABARAP family, and structural studies indicate that it displays a dynamic behavior, participating in a conformational exchange [55,56,57]. Consequently, this structural adaptation of Atg8/LC3/GABARAP is reflected in a set of new functions not observed for other UBLs. For instance, the *N*-terminal α-helices are essential for tubulin binding and oligomerization [55], tethering of lipid bilayers upon autophagosome maturation [58,59], and recognition of mitochondrial phospholipids [60].

Despite their flexibility, *N*-terminal α-helices are specifically aligned to the ubiquitin-like core, forming a deep hydrophobic pocket 1 (HP1, also termed W-site) together with residues of β-strand β2 (Figure 1E, left plot). This pocket binds preferentially indole-based substances, albeit with low affinity [61], and usually accommodates large sidechains of non-polar aromatic residues within the LIRs. HP1 is formed by residues D19, I23, P32, I34, K51, L53, and F108 in LC3B [27]. Another hydrophobic pocket, HP2 (L-site), is built by the solely hydrophobic residues of central α-helix α3 and β-strand β2 (F52, V54, P55, L63, I66, and I67). These two pockets form the so-called *LIR-docking site* (LDS) and mediate a vast majority of known-to-date interactions between SARs, adaptor, and scaffolding proteins with Atg8/LC3/GABARAPs. Of note, LDS occupies the Atg8/LC3/GABARAP surface on an opposite side of the well-known hydrophobic patch (L8-I44-V70) of ubiquitin [62]. On the other hand, the newly described hydrophobic patch on Agt8/LC3/GABARAP surfaces and called the *UIM-docking site* (UDS, Figure 1E, right plot) is similar to the L8-I44-V70 one on ubiquitin and is used by components of the UPS machinery (such as RPN10) and during intracellular trafficking (such as Ataxin-3 and EPS15) [63].

### 2.2. Lessons from Alignment of Atg8 Family Members

Atg8, the single autophagy modifier in yeast, gave the name to Atg8-family proteins. The number of autophagy modifiers in other organisms varies significantly, with strong expansion over the higher metazoans and plants (up to 22 Atg8 family members in some plant species). In humans, there are six Atg8 orthologs: LC3A, LC3B, LC3C (encoded by respective *MAP1LC3*, microtubule associated protein light chain 3, alpha, beta, and gamma genes); and GABARAP (γ-aminobutyric acid receptor-associated proteins), GABARAPL1 (GABARAP-like protein 1) and GABARAPL2 (also known as GATE-16). Despite the fact that LC3C and GABARAPL2 are branched into separate clades on the phylogenetic tree [64,65], human Atg8-family proteins can be broadly categorized into two groups: LC3s and GABARAPs. The sequence alignment (Figure 2) of the Atg8-family members from five different species reveals a clear similarity for the proteins within individual subfamilies. It also indicates that the proteins from the GABARAP subfamily are more evolutionary related to the Atg8 proteins than those of the LC3 subfamily. However, there are substantial differences in the sequences not only between the subfamilies but also between the individual subfamily members. This was proposed to lead to a functional segregation of the LC3 and GABARAP proteins. Indeed, the LC3 and GABARAP proteins were first identified in different compartments of human cells (microtubules for LC3 [66] and synaptic membranes for GABARAP [67]), suggesting different functions for each subfamily. It could subsequently be shown that, upon starvation-induced autophagy, LC3-subfamily proteins are responsible for the elongation of the autophagosomal membranes, while GABARAPs are acting downstream, participating in the maturation and closure steps of the autophagosome formation [68]. Recent studies showed that only GABARAP-subfamily members are important for the activation of the phagophore-priming ULK1-ATG13-FIP200 complex [69,70]. The centriolar satellites protein PCM1 binds unconjugated GABARAP and LC3C proteins to mediate their localization at the pericentriolar material and control autophagic degradation of centriolar satellites and GABARAPs [71,72]. Another example of the selective function of individual LC3/GABARAPs is the recruitment of LC3C to invading bacteria (*Salmonella typhimurium*) via the specific SAR, NDP52, important for autophagy-mediated restriction of the bacterial growth. Depletion of both (NDP52 and LC3C) proteins is followed by an inability of the cell to defend the cytosol against invasion by *S. typhimurium*, while depletion of all other LC3/GABARAPs does not affect it [73]. Investigation of the molecular mechanisms behind these selective functions (e.g., linkage between residues at specific positions within LC3/GABARAP proteins and their functions) is only the beginning.

In plants, the number of Atg8 orthologs varies from 1 in algae to 22 in angiosperms [74]; however, the diversity of plant Atg8 proteins could be significantly higher because of multiple gene duplications in order to adapt to various adverse conditions where Atg8 proteins play a crucial role [75]. The plant Atg8 proteins also reveal significant selectivity in interaction with their interaction partners, originated from the sequence difference between Atg8 isoforms in different species [76]. Of note, all the key residues, participating in the HP1, HP2 [27], and UDS [63] are conserved, as well as the key lysine residues facilitating LIR binding: K49 and K51 in LC3B (K46 and K49 in GABARAPs). The K49 performs a gatekeeper function, regulating the entrance of the aromatic residues of canonical LIRs into the HP1 [77]. Interestingly, K49A mutation significantly enhances binding of canonical LIRs to LC3B protein [77,78], while K51A decreased or abolished it.

The *N*-terminal α-helices show significantly less conservation, which agrees with the hypothesis that these helices predetermine the selectivity of the interactions between Atg8 proteins and LIR motifs in SARs [59] and thus should be different in amino acid content for each individual family member. The few conserved residues within these α-helices participate either in folding of Atg8 proteins or in the formation of HP1. As expected, the loop regions are significantly less conserved, the relatively long loops L1, L2, and L3 show almost no identical or similar residues. Most conserved are regions of all β-strands, indicating their pivotal role in Atg8-protein folding and in the formation of HP1 and HP2.

The most significant consequence of this alignment, in our view, is the clear separation of all Atg8 proteins in LC3B and Atg8/GABARAP subtypes based on a few positions within their sequence. The first constant difference between LC3B and Atg8/GABARAP subtypes of proteins is the switch between the intramolecular electrostatic contacts for residues at positions 8 and 47 in Atg8/GABARAP (positions 10 and 50 in LC3B, respectively). For all Atg8/GABARAPs, position 8 is occupied by a negatively charged or polar residues which are able to play a role of hydrogen bond acceptors (Glu, Asp, Gln, Asn, Ser, and Thr), while position 47 is permanently used for electropositive residues (Lys and Arg), serving as the hydrogen bond donors. In contrast, in LC3 subtype, there are electropositive residues (donors) at position 10 and electronegative residues (acceptors) at position 50. These residues come closely to each other and form intramolecular hydrogen bonds or salt bridges to stabilize the Atg8/LC3/GABARAP structure and ensure a proper orientation of the *N*-terminal α-helical subdomain (Figure 3A). Importantly, these residues also form intermolecular contacts to residues in LIR motifs and, therefore, also contribute to the selectivity of Atg8 interactions with other proteins. It was shown recently that the T50 in LC3B is phosphorylated by a number of kinases [79,80]. This phosphorylation is necessary for normal autophagosome-lysosome fusion and clearance of invading bacteria [80]; inhibiting, however, the autophagic degradation of p62/SQSTM1 [79]. Both studies emphasize significant functional differences as a consequence of the different content at positions 47 (in GABARAPs) and 50/56 (in LC3A, B/LC3C). Another difference is constantly shorter long loops in Atg8 and GABARAPs. The loops between β-strands β1 and β2 (L1) and between β-strand β3 and α-helix α4 (L3) display no conservation; however, they undergo significant dynamic modulations in the free and LIR-bound forms of Atg8/LC3/GABARAPs, as was observed by NMR experiments [34,81]. Therefore, lack of one residue could in principle affect their dynamics and thus modulate selectivity to a specific LIR [50].

Besides these two constant differences, reflecting global separation of LC3 and GABARAP/Atg8 subtypes, there are a several organism-specific hotspots, regulating the difference in the recognition of LIR motifs by the subfamily members. For example, Y25 (invariant in all GABARAPs) participates frequently in the formation of intermolecular hydrogen bonds with positively charged or polar residues at position X_2_ of LIR motifs. The favorable conformation of Y25 is stabilized via cation-π interactions (reviewed in [82]) with a guanidinium moiety of invariant R28 (Figure 3B). The distinct conformation of Y25 and R28 making the intermolecular hydrogen bonds more energetically favorable and thus increasing the affinity of the GABARAP: LIR binding. In LC3, there are H27/H27/F34 (LC3A/LC3B/LC3C) at position Y25, and, therefore, hydrogen bonds could not be formed in the same way, as well as the conformation of aromatic rings could not be stabilized by cation-π interactions with K30/K30/K36. Thus, this set of residues in GABARAPs predetermines the selectivity of LIRs in KBTBD6/7 [83] and PCM1 [71,84] binding to GABARAP proteins. R28 itself forms a hydrogen bond with the negatively charged or polar residues at LIR position X_3_, and cation-π interactions also play a favorable role for the elevated affinity to GABARAPs [34].

Also, post-translational modifications (PTMs) of Atg8/LC3/GABARAPs proteins may affect the affinity and selectivity of LIR:Atg8/LC3/GABARAP interactions. Phosphorylation of Atg8/LC3/GABARAPs at accessible S/T residues has been shown to regulate the selective autophagy [79,80,85,86,87]. Enzymatic acetylation/deacetylation of K49 and K51 regulates LC3 nuclear-cytoplasmic shuttling [88] but might equally regulate the affinity of the binding between LC3 and the LIRs within SARs, as these residues are key elements of the LIR:Atg8/LC3/GABARAP interface. Ubiquitination is another type of PTM modulating autophagy in an LC3- or GABARAP-selective manner. Monoubiquitination of LC3 (and not GABARAPs) at K51 driven by the coordinated action of ubiquitin-activating enzyme UBA6 and the hybrid ubiquitin-conjugating enzyme/ubiquitin ligase BIRC6 targets LC3 for the proteasomal degradation and negatively regulates autophagy [89]. Mib1-driven mono- and polyubiquitination of the GABARAP (not LC3) on K13 and K23 within *N*-terminal α-helical subdomain occurs through K48-chains and targets GABARAPs to proteasomal degradation [72].

## 3. Interactions between Atg8/LC3/GABARAP Proteins and Their Binding Partners

### 3.1. The LIR Concept

The LIR in mammals and the AIM in yeast were described in pioneering biochemical [28] and structural works [26,27] as short polypeptide sequences containing ~10–20 residues. Early structural studies revealed that the core LIR/AIM (LIR thereafter) sequence contains a W-X-X-L motif (where X is any residue). The LIR polypeptide of p62/SQSTM1, for instance, adopts a β-stranded conformation, forming an intermolecular parallel β-sheet with the β-strand β2 of LC3B, while the sidechains of W and L residues occupy the HP1 and HP2 of LC3B, respectively, stabilizing the complex (Figure 4A). Extensive studies in past years provided a more general core consensus, which can be described as Θ-X-X-Γ, where Θ is an aromatic (W/F/Y) and Γ is a hydrophobic (L/I/V) residue (Figure 4B). Investigations of the residues which could occupy the Θ and Γ positions (ether by analyzing the sequences of the hitherto known canonical LIR motifs [25] or by mutational 2D peptide arrays [32,84,90,91] revealed a very high conservation of the three aromatic residues in Θ. As expected from the hydrophobicity profile of HP1, a much higher abundance of solely non-polar Trp and Phe was observed in the native canonical LIRs. In contrast, partially polar Tyr residues are found in only a minority of canonical LIR motifs. It seems that Trp is the most energetically favorable residue for the Θ position. Mutation of the Tyr732 to a Trp increases the NBR1 LIR affinity to GABARAPL1 eight-folds, while the Y732F mutant showed the same affinity [92]. The Phe-containing OPTN LIR shows an eight-fold increase in affinity to LC3B when Phe178 is substituted to a Trp [81]. Of note, the lower affinity of Tyr- and Phe-containing canonical LIR motifs in both aforementioned cases might be associated with the ability of NBR1 and OPTN to regulate autophagic functions [81,92]. The Γ position is a bit less conserved and tolerates large hydrophobic residues, including canonical L/V/I and aromatic residues, except His. Apparently, smaller hydrophobic residues, such as Ala, Pro or Met, do not have enough volume to fill the HP2, while aromatic residues are too big to be docked.

A track of negatively charged residues (Glu/Asp) prior to the Θ enhances the affinity of the LIR interactions with Atg8/LC3/GABARAP. Phosphorylation of residues within the *N*-terminal flanking region of the core LIR (especially directly prior to the aromatic residue) may enhance the affinity of the SAR: Atg8/LC3/GABARAP binding [78,81,93,94,95,96] and serves as a key autophagy regulator in corresponding types of selective autophagy. Phosphorylation of the more distant residues preceding Θ also increases the affinity, however, not so strongly as at positions −3, −2, and −1. In optineurin’s LIR, phosphorylation of individual Ser residues up to position −9 still increases its affinity to LC3B [81]. IKKa-mediated phosphorylation of AMBRA1 S1014 at position −6 promotes AMBRA1’s binding to LC3 and GABARAP (in vitro and in vivo) and serves as a positive regulator of AMBRA1-mediated mitophagy [93]. In some cases, direct phosphorylation of the Tyr residue at the LIR Θ position leads to a weakening of the LIR:LC3/GABARAP binding affinity [94]. Additionally, phosphomimetic mutations in the LIR proximity, for a large number of investigated proteins, increase their affinity to Atg8/LC3/GABARAPs. Interestingly, phosphorylation/phosphomimicking differently affects LIR-dependent interactions with individual members of LC3 and GABARAP subfamilies, indicating that this PTM affects not only the affinity but also the selectivity of the interactions. For example, Beclin1 S93E and S96E mutations enhance binding of Beclin1 to GABARAP and GABARAPL1 three-fold, to LC3A five-fold, and to LC3C eight-fold [91]. Mutational analysis has further shown that certain LIR core sequences possess an increased affinity to the LC3 vs. GABARAP subfamilies of UBLs. This allowed to define a broad consensus for the GABARAP-interacting motif (GIM), conforming to the core sequence (W/F)-(V/I)-X-V [34].

### 3.2. Affinity and Selectivity of Interactions between Atg8/LC3/GABARAPs and LIR Motifs

Affinity (expressed in K_D_ values) of interactions involving Atg8/LC3/GABARAPs is not so high in comparison with affinities of the strongest biointeractions (K_D_~10^−6^ nM for streptavidin:biotin binding [97], up to 10^−3^ nM for antigen:antibody [98]). The K_D_ values for interactions between the canonical p62/SQSTM1 LIR motif and all six human Atg8 proteins were measured by the isothermal titration calorimetry (ITC) experiments and are around 1 µM without any selectivity [92]. Still, these values are >100 times lower in comparison to the interactions between ubiquitin and UBDs (~100 µM, reviewed in [99]). Selectivity, however, cannot be expressed in terms of only one interaction. Generally, selectivity is a comparison between K_D_ values of the two (or more) binding processes. The higher the difference between two K_D_ values, the more selective is the binding. In case of interactions between Atg8/LC3/GABARAP proteins with their partners, we assume that affinity was calculated as K_D_ values for at least one representative member of LC3 and one representative member of GABARAP proteins under the same conditions and by the same method. This may allow us to compare the K_D_ values for them and state if selective interactions take place or not. Table 1 summarizes the most exemplary K_D_ values reported for the interactions between Atg8/LC3/GABARAP proteins and corresponding binding partners.

### 3.3. Atg8/LC3/GABARAP Interactions: LIR and Beyond

During the last 5 years, extensive studies aimed at linking specific proteins with the autophagy pathway were undertaken. As a result of these efforts, ample knowledge on interactions between Atg8/LC3/GABARAPs and various proteins (e.g., SARs, adaptors, targets, and scaffolding factors) at thermodynamic, structural, and functional levels has accumulated and been rationally analyzed. While a majority of the interactions confirmed and refined the concept of the canonical LIR motif, a number of unusual interaction mechanisms were identified outside the LIR concept. Their existence hints at the fact that Atg8/LC3/GABARAPs could provide a significantly broader platform for binding and cellular functions.

Several major types of LIR motifs are used in functionally relevant interactions (Figure 4C–H):

#### 3.3.1. Linear LIR-Like Motifs

The canonical LIR, as well as its variations with several types of the so-called non-canonical sequences, form the most populated group of linear LIR (Figure 4C). They are linear polypeptides from mostly unstructured protein regions with a tendency to adopt a β-stranded conformation. Discovery of a new canonical LIR is facilitated by number of programs such as iLIR [29,30] and hfAIM [101] which analyze the core 4 residue sequence Θ-X-X-Γ as well as it *N*- and *C*-terminal extensions. Unfortunately, the application of the programs is still limited to canonical LIR sequences and does not predict the non-canonical LIRs. Sequences of non-canonical LIRs are characterized by a lack of conserved aromatic residues Θ (as in the cLIR of NDP52 [73]) or an unusual spacing of the aromatic and aliphatic residues, such as that in the combined LIR/UFIM (LIR and UFM1-interacting motif) and its variations, which binds both LC3/GABARAP and UFM1 protein [50,102]. In case of cLIR from the major SAR for mitophagy and xenophagy, NDP52, with core sequence ^133^I-L-V-V^136^ (Figure 4B), HP1 is not engaged by an aromatic sidechain but is rather conformationally adopted to form hydrophobic contacts with the I and L of the cLIR. This interaction is selective for the LC3C protein; the change of I to W increases its affinity but also results in the loss of selectivity [73]. Similar adaptation mechanism is observed for the LIR of the NDP52 homolog, TAX1BP1, which has the core sequence ^140^M-L-V-V^143^ and mediates interactions of this SAR with both LC3 and GABARAP subfamilies [103].

The LIR/UFIM of E1 enzyme for the UBL UFM1, UBA5, with the longer core sequence ^341^W-G-I-E-L-V^346^ exploits hydrophobic contacts of I, L, and V to occupy both HP1 and the HP2 similar to the NDP52 cLIR. However, the evolutionary conserved W engages in a new hydrophobic pocket, termed HP0, which is induced by conformational changes within GABARAPs upon the occupation of HP1 and HP2, making the affinity of the LIR/UFIM:GABARAP interactions comparable with the usual affinity for LIRs (~1 µM range). A set of artificial peptides, rationally derived from UBA5 LIR/UFIM sequence, showed a comparable or slightly higher affinity to LC3B and GABARAPL2 proteins, indicating a high potential of this non-canonical LIR sequence to mediate functionally relevant Atg8/LC3/GABARAP interactions [50].

#### 3.3.2. Three-Dimensional (3D) Interacting Regions (3D LIRs)

Besides linear LIRs, the so-called 3D LIR motifs (some researchers suggest *through-space* or *interspaced LIR* terms) have been discovered (Figure 4D). They mimic the structural arrangement of a canonical LIR by presenting aromatic, aliphatic and negatively charged residues distant in the sequence but kept together in space by the protein’s 3D structure. One such example is the 3D LIR in Atg12 [104], deduced from the structural correspondence of surface-located W185 and I111 in the yeast Atg12~Atg5 complex to W and L in the canonical Atg32 LIR. Mutational analysis confirmed that Atg12 residues W185 and I111 are indispensable for Atg8 membrane recruitment. A 3D-type interaction is also observed between the short *C*-terminal tail of Atg7 (residues 611-630) and Atg8. Spaced in sequence, F619 and I629 occupy in the 3D structure positions of Θ and Γ of the core LIR, and interact with HP1 and HP2 of Atg8, respectively. The negatively charged residues D617, D624, E625, and E628 form intermolecular ionic contacts to the usual set of R and K residues on Atg8 surface, strengthening the interaction [105].

#### 3.3.3. Extended LIR Motifs: *C*-Terminal α-Helical Extensions

*N*- and *C*-terminal extensions of the canonical LIR motifs could dramatically increase affinity of their interactions with the Atg8/LC3/GABARAPs and provide a gain in selectivity. These extensions build additional intermolecular contacts outside the LDS and therefore significantly stabilize the resulting complex. A set of functionally relevant *C*-terminally extended LIRs employ an α-helical structure (Figure 4E), which stabilizes the complex via high-energy intermolecular hydrogen bonds between polar or negatively charged residues within the α-helical extension and the invariant R70 in LC3B (R67 in GABARAPs) [84,106,107,108,109]. This considerable gain in affinity is for example instrumental in selective autophagy of the ER (ERphagy) mediated in *Saccharomyces cerevisiae* by the SAR Atg40. To fragment ER membrane and load into the autophagosome, multiple membrane-resident Atg40 proteins simultaneously engage a number of Atg8 proteins, which they bind with their AIMs tightly, owing to the short helix *C*-terminal to AIMs. Importantly, this feature is conserved all the way to the mammalian ERphagy-specific SARs [110]. Increase in the affinity in some cases can reach >1000 fold. For example, α-helically extended LIR in ankyrins AnkB shows an affinity to GABARAPs in sub-nanomolar ranges (K_D_ values 0.21–0.29 nM) [100]. The physiological importance of the GABARAP:AnkG interaction is underscored by the study in which a LIR-deficient form of AnkG (W1989R), which fails to bind to GABARAP, leads to a massive reduction in forebrain GABAergic synapses associated with hyperexcitability of pyramidal cell and disruptions in the synchronization of neuronal networks [111]. Interestingly, LC3 proteins display a lower affinity for this LIR (K_D_ 3.2–10.5 nM), highlighting the functional specialization within the LC3/GABARAP family. The extended LIR in the mammalian ERphagy SAR, FAM134B, reveals an elevated affinity (K_D_~25 nM), while the same LIR without a *C*-terminal α-helical extension shows usual affinity in sub-micromolar range (K_D_~700 nM) [100].

*C*-terminally α-helical-extended LIRs are not the only type of the extended LIR motifs. The ATG13, PCM1, and ULK1 use linear polypeptide as the *C*-terminal extension of the LIR motif for selective GABARAP binding. They exploit non-polar extensions of HP2 over the central helix α3 in GABARAPs, which is more polar and even charged in LC3 as well as described above intermolecular hydrogen bonds to GABARAP R28 stabilized by cation-π interactions.

#### 3.3.4. Extended LIR Motifs: *N*- and *C*-Terminal β-Stranded Extensions

Another type of *C*- and *N*-terminal LIR extensions, affecting affinity and selectivity of the LIR: Atg8/LC3/GABARAP interactions, are β-strands, which are separated by a loop and form additional β-strands layer in the intermolecular β-sheet (Figure 4F). The *Legionella pneumophila* effector protein RavZ, inhibiting xenophagy, has a tandem LIR motif at its *N*-terminal region (NLIR). Both parts of this tandem LIR have a canonical sequence (^16^F-E-E-L^19^ and ^29^F-D-L-L^32^) and negatively charged residues prior to both F16 and F32; however, only the first LIR in the tandem provides sidechains of F16 and L19 to engage HP1 and HP2. The second part contributes in stabilization of LC3/GABARAP: NLIR complexes with intermolecular ionic interactions mediated by D30, and with a number of hydrophobic contacts mediated by L31 and L32. In this case, H27 in LC3A and LC3B and invariant Y25 in GABARAPs are the key residue in selectivity of interactions [112].

An *N*-terminal β-stranded extension was observed in the LIR of the autophagy-linked FYVE protein (ALFY), which interacts selectively with GABARAP proteins and LC3C, contributing to autophagic clearance of aggregated proteins. ALFY has the canonical LIR motif (^3346^F-I-F-V^3349^), whose binding to GABARAPs is stabilized by additional intermolecular hydrogen bonds between ALFY’s residues E3342, D3344, and Y3351, and GABARAP residues Y25, K46, and D54, respectively. E3342 and D3344 are fixed in a favorable conformation within the additional β-strand preceding a canonical LIR motif. Invariant GABARAP residues K24/Y25/D54 are the selectivity determinants of binding. These are similar to residues K32/F33/E63 in LC3C and different to Q26/H27/H57 in LC3A/LC3B proteins, and mutations Q26K/H27Y/H57D in LC3B drastically enhance its interaction with ALFY [113].

#### 3.3.5. α-Helical Interacting Regions

α-Helical structures could be implemented by proteins to bind Atg8/LC3/GABARAPs (Figure 4G). The α-helical coiled-coil segment of the retroviral restriction factor TRIM5α mediates a direct interaction between TRIM5α and LC3B, allowing simultaneous TRIM5α dimerization and interaction with two LC3B moieties. Besides TRIM5α’s W196 and Q203, whose sidechains are located in the close proximity to the HP1 and HP2, respectively, there are additional close polar and hydrophobic contacts over the LDS stabilizing the complex. However, absence of high-energy intermolecular hydrogen bonds, which appear upon the formation of the intermolecular β-sheet make affinity of TRIM5α:LC3B interaction very low (K_D_~100 µM) [114]. A similar mechanism was shown for the interaction between GABARAP and the anti-apoptotic protein Bcl-2 [115]. In the GABARAP:Bcl-2 complex structure (modelled on the NMR-titration experiment, α-helical segment of Bcl-2 (residues 10-34) covers the LDS surface of GABARAP, accommodating Bcl-2 W30 into the HP1. In addition, a recent study indicates that the *Plasmodium berghei* transmembrane protein UIS3, which blocks the processive LC3/GABARAP interactions with autophagic machinery and facilitates *Plasmodium*’s escape from the autophagic degradation cycle, also interacts with the LDS of LC3/GABARAP in an α-helical mode with K_D_ of 0.24 µM. However, UIS3 may use another pattern of residues for this interaction [116].

#### 3.3.6. UIM-Like Interacting Regions

A conceptually new mechanism of protein–protein interactions for Atg8/LC3/GABARAP was discovered very recently [63]. It was shown that Atg8/LC3/GABARAPs possess the evolutionary conserved hydrophobic patch (referred to as UDS) around F79 and L81 in LC3A/B (I77 and V79 in GABARAPS), capable of accommodating amphiphilic α-helical UIM domains with an affinity in the lower micromolar range (Figure 4H). It was shown that 19 UIM-containing proteins could interact with Atg8s in *A. thaliana* in a UDS-dependent manner. In human, 6 out of 28 tested UIM-containing proteins (EPN1, EPN2, EPN3, Rabenosyn, ATXN3, and ATXN3L) interacted with both LC3 and GABARAP subfamilies via UDS binding with a different degree of selectivity [63]. The discovery of the alternative binding site on Atg8/LC3/GABARAP surface will lead to the identification of new selective autophagy receptors, adaptors and scaffolding proteins in the near future. Besides UIMs, there are more α-helical substructures which could interact with the human LC3 and GABARAP proteins in UDS-dependent way, and the future investigations should be focused on the identification of a complete set of UDS-dependent interactors and on a more detailed structural characterization of this binding type.

## 4. Emerging Atg8/LC3/GABARAP Interaction Motifs and Elements

It was reported that the LC3/GABARAP interactome in human contains ~400 potential candidates under basal autophagy conditions [117]. Only a small number of these proteins were validated and characterized as LC3/GABARAP binders, while validation of the rest and/or discovery of new candidates is complicated by the fact that researchers are looking for more conventional and better characterized canonical LIR motifs as the interaction determinant. This strategy, however, will not work in the light of the latest finding described above. The progress in identification of the different mechanisms for interactions of Atg8/LC3/GABARAPs allows us to suggest a number of possible, still not identified, structural motifs which could be implicated in these interactions and may serve as a starting point for new investigations.

### 4.1. The Anti-Parallel Intermolecular β-Sheet

For all the canonical and non-canonical LIR sequences identified to date, the orientation of the extended β-stranded conformation of the LIR peptide is parallel to the β-strand β2 in Atg8/LC3/GABARAP. The only exception was reported so far is the structure of an artificial peptide K1 in complex with GABARAP [118]. This peptide adopts a complex conformation with N- and C-termini representing an extended β-strand, connected with a 3.10 helix in the middle. The HP1 and HP2 of GABARAP are both occupied by Trp (W11 and W6, respectively), making the situation more “non-canonical.” Nevertheless, one can predict that antiparallel β-stranded linear peptides (Figure 5A) with a reverse order of residues for Θ and Γ positions and with a corresponding *C*-terminal track of negatively charged residues after the core aromatic residue (Γ-X-X-Θ-X^−^-X^−^-X^−^ instead of X^−^-X^−^-X^−^-Θ-X-X-Γ) could efficiently bind Atg8/LC3/GABARAP proteins. Considering another non-canonical linear LIR sequences (cLIR in NDP52 and LIR/UFIM in UBA5), one can predict existence of a very high amount of LIR-like sequences representing this category. The attempt to generate (by a phage display) highly affinitive and highly selective synthetic peptides capable of binding individual members of LC3 and GABARAP subfamilies in human cells led to the generation of a number of sensor molecules; however, they all seem to contain canonical LIR motifs [119].

### 4.2. N-Terminal α-Helical Subdomain Displacement

The *N*-terminal α-helical subdomain in Atg8/LC3/GABARAP proteins is a key evolutionary addition to the core ubiquitin-like fold to separate structurally and functionally the autophagy modifiers from any other UBLs. The α-helices show a significant conformational exchange [55,81,120] and could potentially be separated from the ubiquitin core as the truncated LC3B and GABARAPL2 proteins were still able to perform some functions, like membrane fusion [59]. The first α-helix in LC3B and GABARAPL2 was successfully swapped to emphasize their role in p62/SQSTM1 recognition [121]. Moreover, it was shown that GABARAPL1 being truncated *N*-terminally for the α-helical subdomain and the β-strand β1 could still recognize and bind a number of cognate receptors and proteins (γ2 subunit of GABA_A_ receptor [67], human κ opioid receptor [122], gephyrin [123]).

Based on these facts one can predict that the α-helical subdomain can be displaced from the ubiquitin core of Atg8/LC3/GABARAP proteins by another α-helical structure (Figure 5B) containing a combination of residues, which are more favorable for the binding of the ubiquitin core of a particular Atg8/LC3/GABARAP protein. More aggressive conditions, which appear in close proximity to membranous structures or in cellular compartments with critical pH values, might facilitate the displacement. In this case, the amino acid content of the displacing α-helices could significantly differ from that for displaced helices α1 and α2, leading to the HP1 modulation in shape and dynamics. That could promote an effective binding of alternative LIR-like sequences from proteins which solely interact with intact Atg8/LC3/GABARAPs.

### 4.3. LIR-Based Atg8/LC3/GABARAP Superbinders or Combinatorial Binder

Interesting applications might be seen from the combination of all possible binding mechanisms in a single polypeptide. Taking into account that the displacing α-helical structure might serve for enhanced selectivity of the polypeptide (according to the role of displaced helices α1 and α2) and that affinity of binding is controlled by canonical or non-canonical LIR core with a *C*-terminal α-helical extension (like in AnkB/AnkG [100]) with a following UIM, one could design a polypeptide which would bind selectively a single member of Atg8/LC3/GABARAP family (Figure 5C). Such a polypeptide will block all the binding sites on Atg8/LC3/GABARAP surface, thus inactivating a specific autophagy modifier in cells. Coupled to a labelling group (GFP, fluorescein, Alexa, etc.), the polypeptide could reveal cellular distribution of a specific autophagy modifier.

## 5. Therapeutic Exploitation of the LIR: Atg8/LC3/GABARAP Interactions

With the growing appreciation of the roles that Atg8/LC3/GABARAP proteins play in the cell, most significantly in various selective autophagy pathways, therapeutic exploitation of these lipid modifiers rapidly comes into focus. It is conceivable that engineering strong and selective binders to individual members of the Atg8/LC3/GABARAP family, on the one hand, and pathological cargo proteins, on the other hand, will enable targeted degradation of complex substrates by the autophagy pathway. This especially may be applicable for multiple neurodegenerative diseases marked by accumulation of pathogenic protein aggregates, such as those containing huntingtin (characteristic of Huntington’s disease), β-amyloid peptide (Alzheimer’s disease), or α-synuclein (Parkinson’s disease) [124]. Such SAR mimetics would have to penetrate the blood-brain barrier and access sick neurons in order to bring about their therapeutic effect. Increased affinity of a peptide or small molecule for LC3/GABARAP could be gained by applying the principles identified in the aforementioned structural studies, e.g., of very strong LIR-based binders [100] or from the emerging combined “superbinders” (Figure 5C). A preclinical proof of principle for such compounds was recently demonstrated for dual-specificity small molecules that were selected to bind mutant huntingtin and LC3B simultaneously. Excitingly, these prototypic selective autophagy inducers could lower levels of mutant huntingtin aggregates in fly and mouse neurons in vivo, showing the potential of this approach [125]. A similar approach may be used to fight infectious diseases, i.e., via enhanced targeting of intracellular pathogens to autophagosomes one could achieve both clearance of the pathogen and its improved presentation to the immune system. Thus, the linear fusion of *Mycobacterium tuberculosis* antigen LpqH in line with LC3B led to enhanced delivery of the protein to autophagosomes and lowered mycobacterial load in immunized mice challenged with a virulent form of *M. tuberculosis* [126].

Inhibition of autophagy is of interest in other disease settings, such as in cancer, where autophagy has been described to provide growth advantage to Ras-driven tumors [127]. Here, design of strong LIR-based Atg8-protein binders may compete for SARs to reduce clearance of different cargo: e.g., accumulation of damaged mitochondria by blocking NDP52- and OPTN-mediated mitophagy may lead to increased apoptosis in cancer cells, in which this suicide pathway is still active. Intriguingly, it was shown that autophagy inhibition also led to the stabilization of MHC-I antigen complexes and enhanced anti-tumor immunity in combination with immune checkpoint inhibitors. A major ubiquitin-binding SAR NBR1 was recently implicated in this process [128]. Autophagy inhibition should be done with significant cautions, as mice in which autophagy was inhibited in an inducible manner (*Atg7* knockout) developed multiple abnormalities involving brain, liver, and muscle and succumbed to infectious diseases within 2–3 months of the onset of the autophagy block [17].

Better understanding of what drives selectivity of LIR peptides to individual members of LC3/GABARAP family can help design more specific treatments in the future. Several groups performed rational design or screening to develop peptides with desirable specificities. These could be used as fluorescent sensors for detecting LC3/GABARAPs in real time, allowing live imaging of autophagosome formation [129] as well as different selective autophagy processes, such as mitophagy and xenophagy [119]. A similar approach can be adopted for screening for peptides with super high affinity toward LC3/GABARAP proteins for potential use in therapeutic applications. Structural insights into the complexes between high-affinity peptide interactors and the LC3/GABARAP core will eventually help develop peptidomimetic compounds with superior pharmacological properties and amenability to oral administration for use in patients with neurodegenerative and infectious diseases as well as cancer.

## Figures and Tables

**Figure 1 cells-09-02008-f001:**
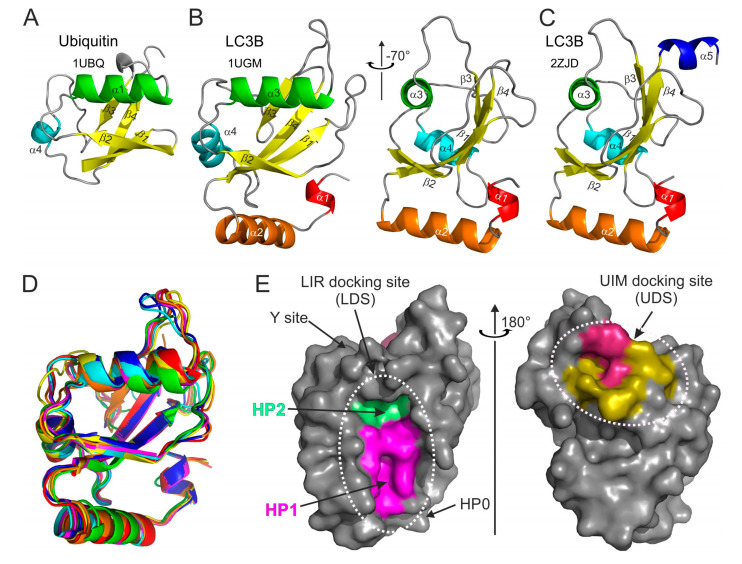
Structural features of ubiquitin and Atg8/LC3/GABARAP proteins. Ribbon diagrams of the (**A**) ubiquitin and (**B**) LC3B structures aligned to the common ubiquitin core (left plot in **B**). The right plot was generated by rotation of the LC3B structure by −70° around the *Y*-axis. The secondary structure elements in both proteins are colored in rainbow color-code from α-helix α1 in LC3B (red-orange-green-cyan-blue), all β-strands are colored yellow. The PDB ID codes of each structure are presented under the protein names. (**C**) LC3B structure with *C*-terminal α-helix α5 (the same orientation as the right plot in (B)). (**D**) Structural alignment of yeast Atg8 and human LC3A, LC3B, LC3C, GABARAP, GABARAPL1, and GABARAPL2 (rainbow color-code) proteins shown as ribbon diagrams (the same orientation as the left plot in (**B**)). (**E**) Left plot: surface representation of LC3B structure (the same orientation as the left plot in (**B**)), showing the main interacting sites—HP1 (magenta) and HP2 (light green), which form the LC3 docking site (LDS). Position of additional interacting sites, like HP0 [50], Y-site [51], etc., are indicated by arrows. The alternative interacting area, the UIM docking site (UDS), is located on the opposite side of the LC3B molecule (right plot). The most relevant residues are colored dark red, additional hydrophobic residues around it are colored yellow.

**Figure 2 cells-09-02008-f002:**
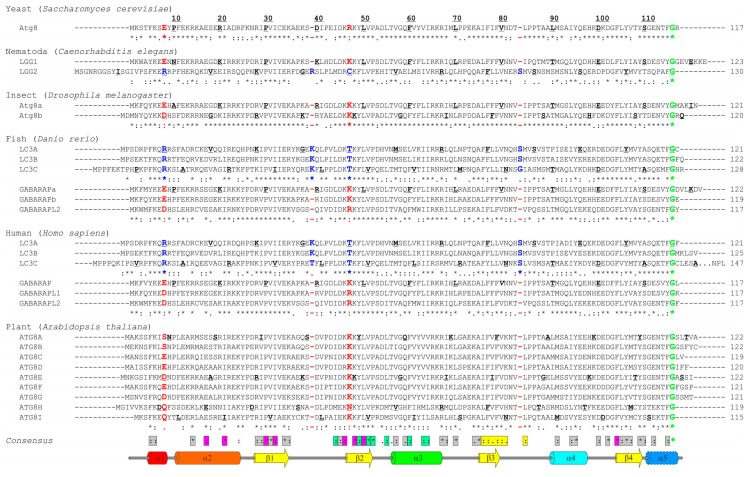
Sequence alignment of Atg8/LC3/GABARAP proteins. Sequence alignment of the Atg8-family members from six model species—yeast (*Saccharomyces cerevisiae)*, Nematoda (*Caenorhabditis elegans*), insect (*Drosophila melanogaster*), fish (*Danio rerio*), human (*Homo sapiens*), and plant (*Arabidopsis thaliana*). Secondary structure elements from the human LC3B (PDB ID 2ZJD) are shown on top (color-code as in Figure 1B,C). Every tenth residue in each sequence is marked bold/underlined, the catalytic Gly is marked green. The identity scores (asterix, * , for identical residues; colon, : , for very similar residues; dot, . , for analogous residues; space, , for residues without any similarity; dash, - , for gaps) are presented below each group of the Atg8/LC3/GABARAP. The residues (or their absence) separating GABARAP/Atg8 and LC3 protein subtypes are marked red and blue, respectively. The consensus string for all 28 proteins is presented at the bottom of alignment. The residues showed conservation are grouped within the following classes: residues participating in the protein folding (grey); residues forming HP1 (magenta); residues forming HP2 (light green); and residues forming UDS (yellow).

**Figure 3 cells-09-02008-f003:**
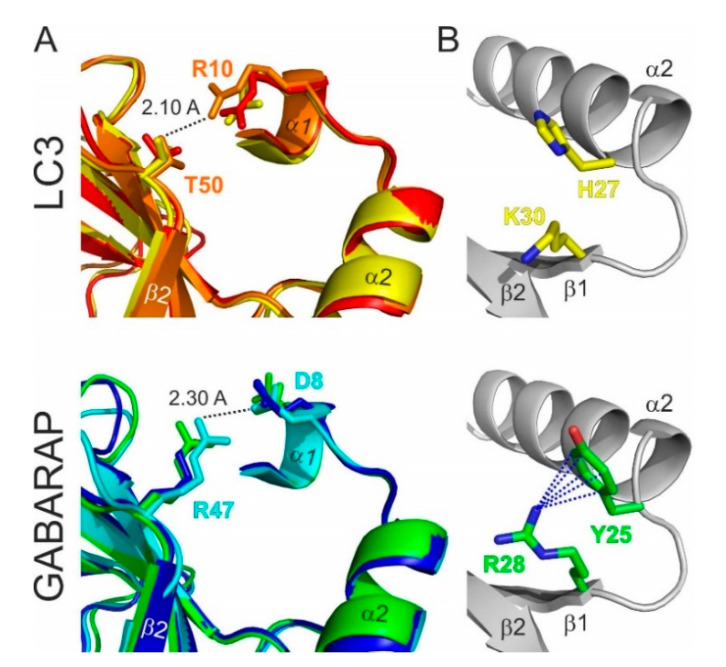
Structural differences between the LC3 and GABARAP proteins. (**A**) Specific structural difference between GABARAP/Atg8 and LC3 protein subtypes. Intramolecular contacts within LC3 (top) and GABARAP (bottom) proteins; color code as in Figure 1D. Involved residues are presented as sticks; the shortest distance is given for specific protein residues (indicated at the plot). (**B**) Orientation of H27 and K30 sidechains in LC3B (top) and corresponding Y25 and R28 sidechains in GABARAPs (bottom). Cation-π interactions (the non-covalent electrostatic interaction between an electron-reach face of aromatic rings and adjacent cations), stabilizing the specific orientation of Y25/R28 sidechains in GABARAPs are shown as dashed lines.

**Figure 4 cells-09-02008-f004:**
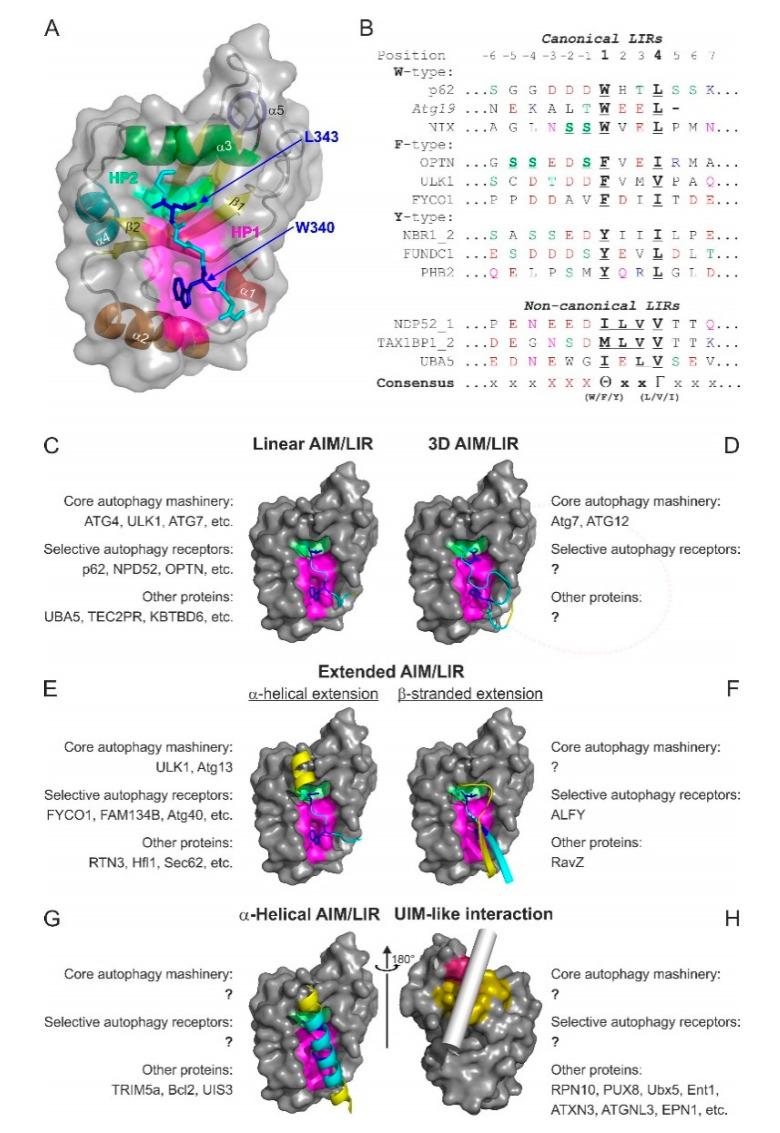
Atg8/LC3/GABARAP interactions with their partners. (**A**,**B**) The LIR concept. Structure of p62/SQSTM1-LIR:LC3B complex (PDB ID 2ZJD) (**A**). LC3B is shown as a semi-transparent surface with the structural elements (α-helices and β-strands) visible. p62/SQSTM1-LIR is shown as a main chain (cyan) with sidechains of core LIR residues (W340 and L343, blue) as sticks. Two hydrophobic pockets of LC3B, accommodating W340 and L343 sidechains, are shown on LC3B surface (HP1—magenta, HP2—light green). (**B**) Alignment of canonical (upper section) and non-canonical (lower section) LIR motifs with positions of residues indicated on top (from −6 to +7). Negatively charged residues (red), polar residues (orange), and phosphorylatable residues (green) are indicated over the LIR sequences. The phosphorylatable residues confirmed to be phosphorylated are marked bold/underlined. The underlined characters within core LIR sequences indicate residues whose sidechains are accommodated by HP1 and HP2. (**C**–**H**) Types of Atg8/LC3/GABARAP-interacting motifs and elements are shown as examples of known structures. For all plots, LC3B surface in orientations as in Figure 1E is shown. The interacting elements are given as ribbon diagrams for the known structures; in the case of Ubx5, the putative position of the helical ubiquitin-interacting motif (UIM) is indicated by a gray cylinder. Within the interacting elements, residues contributed with sidechains into HP1 and HP2 are shown in sticks and colored blue, residues within the close LDS contacts are colored cyan, and residues with other contacts are yellow. For each type, the names of the known interactors are indicated, as well as their functional roles sorted in groups of “core autophagy machinery,” “selective autophagy receptors,” and “other proteins.”

**Figure 5 cells-09-02008-f005:**
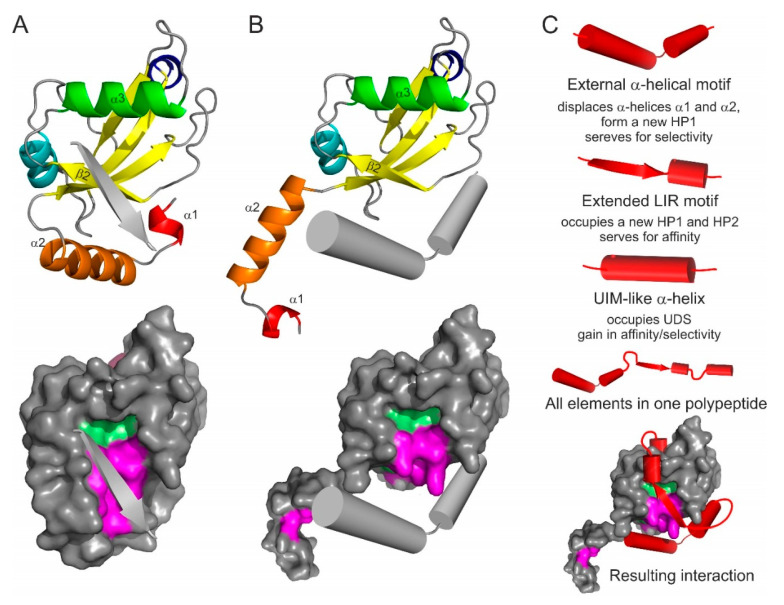
Emerging types of Atg8/LC3/GABARAP interacting motifs and elements: the antiparallel β-strand (**A**), and displacing α-helical structure (**B**). The interacting elements shown as grey arrows or cylinders on LC3B ribbon diagram (top) and on LC3B surface (bottom, with HP1 and HP2 indicated). (**C**) Superbinder construction. Combination of the known/hypothetical binding elements (shown as red cylinders/arrows for each type), complete polypeptide with all the elements, and the resulting position of the “superbinder” on the surface of LC3B.

**Table 1 cells-09-02008-t001:** K_D_ values measured for interactions between different LIRs and LC3/GABARAP proteins. All K_D_ values are given in µM.

Protein	LC3A	LC3B	LC3C	GABARAP	GABARAPL1	GABARAPL2	Method	Ref.
AnkG	0.55	0.34	2.39	2.6 × 10^−3^	3.7 × 10^−3^	40 × 10^−3^	ITC	[100]
AnkB	3.7 × 10^−3^	4.2 × 10^−3^	10.5 × 10^−3^	0.27 × 10^−3^	0.29 × 10^−3^	0.21 × 10^−3^	ITC	[100]
PLEKHM1	4.22	6.33	3.45	0.55	0.77	0.93	ITC	[34]
PCM1	292	982	17.9	2.0	1.6	14.4	BLI	[84]
ULK1	5.9	48.2	2.5	50 × 10^−3^	48 × 10^−3^	0.53	BLI	[84]
ATG13	4.1	9.6	0.48	0.59	0.53	3.1	BLI	[84]
FIP200	281	1206	63.3	5.6	7.0	86.4	BLI	[84]
p62/SQSTM1	2.0	4.5	2.7	0.9	0.6	5.2	BLI	[84]
Ambra1	>100	>100	>100	39	>50	>100	ITC	[93]
pS^1014^Ambra1	(50)	(50)	(>100)	(21)	(25)	(>100)		

ITC—isothermal titration calorimetry, BLI—bio-layer interferometry.

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
