# Peer review of "Atg8-Family Proteins—Structural Features and Molecular Interactions in Autophagy and Beyond"

_cells, 2020, doi:10.3390/cells9092008_

Round 1
Reviewer 1 Report
The manuscript by Wesch et al is a very comprehensive analysis of the structural features and interactions of key components of autophagic pathway - ATG8 family proteins. The review is very well organized, very focused and extremely helpful in classification and understanding of structural determinants of ATG8 family proteins and their interacting partners. To my opinion this review can be published as it is without any essential changes. I only have a couple of suggestions for figures and several minor corrections.
- Would it be possible to put the Figure 1C in the same upper row as Figures 1A and 1B? Because B and C are the same proteins – it will be better to have them nearby. Also, in the figure legend it states that “C” is in the same orientation as right panel “B”, yet beta sheet 4 position is not looking the same. Alternatively, would it be possible to replace 1UGM structure in B by 2ZJD (with appropriate rotation), like that you will have both the alignment with common ubiquitin core AND alpha5 helix in the same figure.
- I would split figure 2 in two figures – one containing alignments (A-C), another – showing structures. Also, if A and B can be shown in the same page, they could probably be merged and marked as one figure panel (which is the case for the current Figure 2C). In the figure legend (lane 250) catalytic Gly is not “highlighted in green”, it is “marked in green”.
- Lane 173 – insert “)” after reference number
- lane 273 – need to have a comma and space between [61] and “Interestingly”
- lanes 310-314. This sentence is too long. The reference to the non-covalent electrostatic interaction in between this sentence (in brackets) makes it even harder to go through this phrase.
- do not need to give the definition for PTMs twice (lane 321 and 373)
- Figure 3H – do not need to write “etc” twice in the list of “other proteins”
- Lane 401 – delete ‘found in biology”
- Lane 404 – change “amounted” to “were” or something else
- Lane 405 – “these values are >100 times more affine” – 100 times higher?
- Lane 515 – 3.3.5. (not 3.3.5.5.)
- “In vivo” and “in vitro” should be in italic
Author Response
Reviewer comments:
We thank all the reviewers for the positive feedback, it is important for us to know that the independent referees found our work of interest for other researchers. Indeed, the reviewers comments and suggestions were very fair and permitted us to improve the quality of our review. We have spent significant time and effort to respond to all the points, and we hope that we could address all of them. Please find below our point-by-point responses.
Reviewer #1:
Comments and Suggestions for Authors
The manuscript by Wesch et al is a very comprehensive analysis of the structural features and interactions of key components of autophagic pathway - ATG8 family proteins. The review is very well organized, very focused and extremely helpful in classification and understanding of structural determinants of ATG8 family proteins and their interacting partners. To my opinion this review can be published as it is without any essential changes. I only have a couple of suggestions for figures and several minor corrections.
- Would it be possible to put the Figure 1C in the same upper row as Figures 1A and 1B? Because B and C are the same proteins – it will be better to have them nearby. Also, in the figure legend it states that “C” is in the same orientation as right panel “B”, yet beta sheet 4 position is not looking the same. Alternatively, would it be possible to replace 1UGM structure in B by 2ZJD (with appropriate rotation), like that you will have both the alignment with common ubiquitin core AND alpha5 helix in the same figure.
Reply: We modified the Figure 1 according to the first suggested option. The difference between position and propensity of β-strand β4 as well as some other small differences seen in the plots are due to the differences within each individual structure. The Figures 1C (old and new) were generated in PyMol program using alignment of the molecule 2ZJD to 1UGM and exporting the ribbon diagrams of 1UGM and 2ZJD in separate graphical files.
- I would split figure 2 in two figures – one containing alignments (A-C), another – showing structures. Also, if A and B can be shown in the same page, they could probably be merged and marked as one figure panel (which is the case for the current Figure 2C). In the figure legend (lane 250) catalytic Gly is not “highlighted in green”, it is “marked in green”.
Reply: We specially thank the reviewer for this comment. The complete sequence of all Atg8-family member in a single plot will obviously increase the understanding of our alignment (however, will require a higher resolution of the published figure). Based on the suggestion, we split the former Figure 2 in two figures - Figure 2 and Figure 3, where Figure 2 consists of solely alignment and Figure 3 represents the known structural features of the LC3 and GABARAP/Atg8 subtypes.
We also modify the Figure 2 legend regarding the catalytic Gly to read “marked green.” (in the line 265 now).
- Lane 173 – insert “)” after reference number
Reply: We inserted a bracket after the reference in the line 173 after reference number to read “the ‘ubiquitin-like’ superfamily (structural classification of proteins database, SCOP [38]).” in the line 196 now.
- lane 273 – need to have a comma and space between [61] and “Interestingly”
Reply: We made corresponding correction (in the line 300 now).
- lanes 310-314. This sentence is too long. The reference to the non-covalent electrostatic interaction in between this sentence (in brackets) makes it even harder to go through this phrase.
Reply: We reformulate the text regarding cation-π interactions (lines 346-350 now) to make it more understandable:
“The favorable conformation of Y25 is stabilized via cation-π interactions (reviewed in [66]) with a guanidinium moiety of invariant R27 (Figure 3B). The distinct conformation of Y25 and R27 making the intermolecular hydrogen bonds more energetically favorable and thus increasing the affinity of the GABARAP:LIR binding.”
The short definition of the cation-π interactions is moved now in the Figure 3 legend (line 339).
- do not need to give the definition for PTMs twice (lane 321 and 373)
Reply: We removed the second PTM definition (in the line 430 now).
- Figure 3H – do not need to write “etc” twice in the list of “other proteins”
Reply: The second “etc.” was removed from the Figure 3H, the list of the known molecules interacting Atg8/LC3/GABARAP proteins via UDS are listed now in the order from original paper of Marshall et al, 2019.
- Lane 401 – delete ‘found in biology”
Reply: Simply deleting the phrase “found in biology” will make the statement too broad and therefore incorrect (some chemical interaction are in KD orders of 10-60 M). Therefore, we changed the statement in the line 439 of the new manuscript version to read:
“in comparison with affinities of the strongest biointeractions (KD ~ 10-6 nM for”
- Lane 404 – change “amounted” to “were” or something else
Reply: We changed the statement in the lines 442-443 to read:
“are around 1 µM without any selectivity [92].”
- Lane 405 – “these values are >100 times more affine” – 100 times higher?
Reply: 100 times lower (indicating higher affinity. We changed the statement correspondingly (line 443 now).
- Lane 515 – 3.3.5. (not 3.3.5.5.) – Corrected (line 556 now).
- “In vivo” and “in vitro” should be in italic - Corrected (line 424 now).

Reviewer 2 Report
This is a comprehensive review on the biochemical and structural characteristics of Atg8 protein family.
One minor suggestion is to include any information regarding the genetic role of different Atg8 protein members, if anything is known about their physiological functions, such as yeast, plants, worms, flies and mice.
Author Response
Reviewer comments:
We thank all the reviewers for the positive feedback, it is important for us to know that the independent referees found our work of interest for other researchers. Indeed, the reviewers comments and suggestions were very fair and permitted us to improve the quality of our review. We have spent significant time and effort to respond to all the points, and we hope that we could address all of them. Please find below our point-by-point responses.
Reviewer #2:
Comments and Suggestions for Authors
This is a comprehensive review on the biochemical and structural characteristics of Atg8 protein family.
One minor suggestion is to include any information regarding the genetic role of different Atg8 protein members, if anything is known about their physiological functions, such as yeast, plants, worms, flies and mice.
Reply: We thank the reviewer for their great suggestion to expand a bit on the physiological functions of Atg8s and the conjugation machinery. In response, we have now added a paragraph that describes major findings in several organisms while citing relevant literature (lines 69-82 in the new version of the manuscript). We hope this additional part will further strengthen this manuscript.

Reviewer 3 Report
The authors provided a review on the current understanding of the structure, the principle of substrate binding, and molecular interaction of Atg8 family proteins. Overall, this manuscript is very detailed and well-organized in a way the readers can easily comprehend. It deals with a topic that will be of interest to the readers in the field of autophagy research. However, some major and minor points should be addressed before the acceptance of the manuscript.
Major points
- (Line 139-141) Why the three UBL moieties in Atg12-Atg5 complex comprises are important for the recruitment of other factors for phagophore elongation? These do not seem to directly relate. The authors need to elaborate or cite a few references.
- (Line 182) Regarding the statement that the C-terminal helix of LC3 could participate in protein-protein interaction outside the autophagy pathways, based on which observation this assumption was made? It would be preferable to provide some explanation or cite papers.
- (Line 442) The author should provide the reference in which the effect of the I-to-W mutation of LC3C was reported.
- The term “3D LIRs” in section 3.3.2 may be confused with the “alpha-helical LIR” in section 3.3.5. Perhaps “Interspaced LIR” will be more fitting.
Minor points
A few typos and errors are listed below:
- Line 116: Affine à Affinitive
- Line 148: Stricture à Structure
- Line 427: Motives à Motifs
- Line 416: K”D” should be subscripted
- Line 637: MHCI- àMHC-I
Author Response
Reviewer comments:
We thank all the reviewers for the positive feedback, it is important for us to know that the independent referees found our work of interest for other researchers. Indeed, the reviewers comments and suggestions were very fair and permitted us to improve the quality of our review. We have spent significant time and effort to respond to all the points, and we hope that we could address all of them. Please find below our point-by-point responses.
Reviewer #3:
Comments and Suggestions for Authors
The authors provided a review on the current understanding of the structure, the principle of substrate binding, and molecular interaction of Atg8 family proteins. Overall, this manuscript is very detailed and well-organized in a way the readers can easily comprehend. It deals with a topic that will be of interest to the readers in the field of autophagy research. However, some major and minor points should be addressed before the acceptance of the manuscript.
Major points
- (Line 139-141) Why the three UBL moieties in Atg12-Atg5 complex comprises are important for the recruitment of other factors for phagophore elongation? These do not seem to directly relate. The authors need to elaborate or cite a few references.
Reply: We thank the reviewer for this important remark and have in response added some additional explanations and relevant references to highlight the importance of the UBLs found within the Atg5-Atg12 complex (lines 156-163 in the new version of the manuscript).
- (Line 182) Regarding the statement that the C-terminal helix of LC3 could participate in protein-protein interaction outside the autophagy pathways, based on which observation this assumption was made? It would be preferable to provide some explanation or cite papers.
Reply: The reviewer touches an important point with this comment. To our knowledge, no ideas on the conditional existence, stability and functional role of the additional helix α5 were reported over ~12 years after the first structure (2ZJD, Ichimura et al, 2008) and more than 10 structures of full-length LC3A, LC3B and GABRAP proteins in complex with various LIRs, displaying presence of the α5. We would like to focus attention of researches on this issue and encourage them to put efforts to identify possible roles of the helix. Based on that, we modified the whole paragraph to provide more explanations and to make it softer (lines 199-210 in the new version of the manuscript).
- (Line 442) The author should provide the reference in which the effect of the I-to-W mutation of LC3C was reported.
Reply: All statements regarding NDP52 cLIR and its interactions with LC3/GABARAP proteins are based on the original publication of on von Muhlinen et al, 2012. We introduced this reference (#72, in line 481 now).
- The term “3D LIRs” in section 3.3.2 may be confused with the “alpha-helical LIR” in section 3.3.5. Perhaps “Interspaced LIR” will be more fitting. – pleased the reviewer, saying “(some researchers suggest “Interspaced LIR)” in the text
Reply: We still think that the term 3D LIR is better describing this specific way of interactions and is significantly distant from the alpha-helical LIRs. To be complete, we indicated that the 3D LIRs can be mentioned as “through-space” or “interspaced” LIRs (lines 495-496 now).
Minor points
A few typos and errors are listed below:
Reply: All the indicated errors/typos were corrected in the new version of the manuscript.
- Line 116: Affine à Affinitive – Corrected (lines 130 and 611 now).
- Line 148: Stricture à Structure – Corrected (lines 170 and 632 now).
- Line 427: Motives à Motifs – Corrected (line 465 now).
- Line 416: K”D” should be subscripted – Corrected (line 454 now).
- Line 637: MHCI- àMHC-I – Corrected (line 678 now).
